# Magnetic quasi-atomic electrons driven reversible structural and magnetic transitions between electride and its hydrides

Seung Yong Lee [1,2,6], Dong Cheol Lim[1,3,6], Md Salman Khan [1,3,6], Jeong Yun Hwang[4], Hyung Sub Kim[5], Kyu Hyung Lee [4] ✉ & Sung Wng Kim [1,3] ✉

In electrides, interstitial anionic electrons (IAEs) in the quantized energy levels at cavities of positively charged lattice framework possess their own magnetic moment and interact with each or surrounding cations, behaving as quasi-atoms and inducing diverse magnetism. Here, we report the reversible structural and magnetic transitions by the substitution of the quasi-atomic IAEs in the ferromagnetic two-dimensional $[Gd_2C]^{2+} \cdot 2e^-$ electride with hydrogens and subsequent dehydrogenation of the canted antiferromagnetic $Gd_2CH_y$ ($y > 2.0$). It is demonstrated that structural and magnetic transitions are strongly coupled by the presence or absence of the magnetic quasi-atomic IAEs and non-magnetic hydrogen anions in the interlayer space, which dominate exchange interactions between out-of-plane Gd–Gd atoms. Furthermore, the magnetic quasi-atomic IAEs are inherently conserved by the hydrogen desorption from the $P\bar{3}1m$ structured $Gd_2CH_y$, restoring the original ferromagnetic state of the $R\bar{3}m$ structured $[Gd_2C]^{2+} \cdot 2e^-$ electride. This variable density of magnetic quasi-atomic IAEs enables the quantum manipulation of floating electron phases on the electride surface.

Interstitial anionic electrons (IAEs) construct an ionic crystal as an essential ingredient together with positively charged lattice framework, forming an electride, which is distinguished from conventional ionic compounds with defect color centers trapping electrons[1–3]. Benefited from the intriguing nature of IAEs, exotic physical and chemical properties of electrides, such as low work function[4], high electronic mobility[5], excellent electron reservoir[6,7], efficient catalytic activity[8,9], and quantum properties of magnetism[10,11], superconductivity[12], and topology[13], have attracted considerable interests in both fundamental science and practical applications. This triggers the exploratory research for the discovery of a uncharted class of electrides over the past decades, leading to the success in finding several two-dimensional (2D) electrides; $[Ca_2N]^+ \cdot e^-$ and $[Re_2C]^{2+} \cdot 2e^-$ (Re = Y, Sc, and Gd) with IAEs at interlayers[5,11,14] and van der Waals $[ReCl]^{2+} \cdot 2e^-$ (Re = Y and La) with IAEs at intralayers[15–17]. Emergent quantum properties of such 2D electrides are governed by the localization degree of IAEs and their hybridization with neighboring cations. In particular, topological Weyl and ferromagnetic states in $[Gd_2C]^{2+} \cdot 2e^-$ electride with strong hybridization of IAEs and Gd cations are of great interest in the field of quantum materials[11,18].

Meanwhile, the magnetism of 2D electrides originates from the existence of strongly localized IAEs at interlayer space, which have own magnetic moments and facilitate the magnetic interaction with neighboring cations[10,11,17]. For example, the IAEs in the $[Y_2C]^{2+} \cdot 2e^-$ play as ferromagnetic particles in the lattice framework composed of only paramagnetic elements, exhibiting the superparamagnetism[10]. The

[1]Department of Energy Science, Sungkyunkwan University, Suwon 16419, Republic of Korea. [2]KIURI Institute, Yonsei University, Seoul 03722, Republic of Korea. [3]Center for Electride Materials, Sungkyunkwan University, Suwon 16419, Republic of Korea. [4]Department of Materials Science and Engineering, Yonsei University, Seoul 03722, Republic of Korea. [5]Neutron Science Division, Korea Atomic Energy Research Institute, Daejeon 34057, Republic of Korea. [6]These authors contributed equally: Seung Yong Lee, Dong Cheol Lim, Md Salman Khan. ✉e-mail: khlee2018@yonsei.ac.kr; kimsungwng@skku.edu

$[Gd_2C]^{2+}\cdot 2e^-$ was found to be the room-temperature ferromagnetic electride with $T_C$ of 350 K due to the exchange interactions of interlayer Gd cations across IAEs[11]. Importantly, the IAEs in $[Gd_2C]^{2+}\cdot 2e^-$ are considered as magnetic quasi-atomic electrons with substantive magnetic moments that are responsible for the occurrence of ferromagnetism from the antiferromagnetic $[Gd_2C]^{2+}$ lattice framework. The concept of interstitial quasi-atomic electrons (IQEs), suggested by Miao and Hoffmann, has been esteemed to understand the nature of elemental and compound electrides on the basis of theoretical ground[19]. Indeed, potassium electride under high pressure is stabilized by ferromagnetic ordering of the IQEs[20]. Furthermore, the mixed-cation $[YGdC]^{2+}\cdot 2e^-$ electride exhibited the ferrimagnetic state, which was attributed to the direct exchange interactions between magnetic IQEs at different crystallographic positions[21]. In addition to the magnetic ordering of IAEs in the electrides, the IAEs on the cleaved surface of 2D $[Gd_2C]^{2+}\cdot 2e^-$ electride are found to be spin-polarized Fermi liquid and crystallized into the hexatic phase by decreasing their density on the surface[22].

The IQEs in the magnetic electrides can be regarded as analogous to the substituted or doped magnetic elements in typical magnetic alloys[23], indicating that the presence of the IQEs can provide a freedom to tune the magnetic properties and study their role in magnetic phase transitions of electrides. In the previous report[11], the substitution of chlorine atoms for the IAEs in ferromagnetic $[Gd_2C]^{2+}\cdot 2e^-$ electride resulted in the transition to antiferromagnetic $Gd_2CCl$ and proved the presence of magnetic IQEs. On the other hand, hydrogen-substituted electrides, in which hydrogens absorb the IQEs and form the hydrogen anions, have been examined to find the crystallographic positions of IQEs and elucidate their contribution to the electronic density of state[24,25]. Recent computational studies suggested that the hydrogenation for the monolayers of 2D $[Ca_2N]^+\cdot e^-$ and $[Gd_2C]^{2+}\cdot 2e^-$ electrides significantly altered their magnetic properties[26,27]. However, an experimental investigation of hydrogenation of the magnetic 2D electrides has been hardly ever reported in spite of a possibility to identify the critical role of magnetic IQEs for triggering the magnetism and tuning the magnetic properties. This might come from the experimental difficulty in handling the chemically unstable electrides as well as a common expectation for the hydrogen-induced embrittlement.

Here, we report the hydrogenation and dehydrogenation of the ferromagnetic $[Gd_2C]^{2+}\cdot 2e^-$ electride, which simultaneously induced the magnetic and structural phase transitions. Depending on the relative concentration between IQEs and hydrogen anions, crystal structure, and magnetic phase exhibited the strongly coupled reversible transitions between the ferromagnetic layered structure of the $R\bar{3}m$ space group to the canted antiferromagnetic layered structure of the $P\bar{3}1m$ space group, providing an experimental proof on the magnetic nature of IQEs.

## Hydrogen-induced phase transition in $[Gd_2C]^{2+}\cdot 2e^-$ electride

The pristine $[Gd_2C]^{2+}\cdot 2e^-$ electride is crystalized in anti-$CdCl_2$-type layered structure belonging to the rhombohedral $R\bar{3}m$ space group, where the IQEs are occupying the interlayer space (Fig. 1a). Note that the occupancy of IQEs can be found in both octahedral and tetrahedral sites of Gd-sublattice at the interlayer space. From the electron localization function (ELF) obtained by theoretical calculations of $[Y_2C]^{2+}\cdot 2e^-$, $[Gd_2C]^{2+}\cdot 2e^-$, and $[YGdC]^{2+}\cdot 2e^-$ electrides[10,11,21], it was revealed that most IQEs occupy the octahedral sites with a minor occupancy at tetrahedral sites. Substitution of IQEs with hydrogens in the isostructural 2D electrides has been experimentally investigated for $[Ca_2N]^+\cdot e^-$ and $[Y_2C]^{2+}\cdot 2e^-$, exhibiting a distinct difference in structural transition[28–30]. Hydrogenation of $[Ca_2N]^+\cdot e^-$ electride with fully delocalized IAEs leads to the transition from 2D layered structure to 3D-like a cubic structure of $Fd\bar{3}m$ (No. 227) space group due to the penetration of excess hydrogens into the positively charged $[Ca_2N]^+$

layers[28]. On the contrary, hydrogens only occupied the interlayer space between positively charged $[Y_2C]^{2+}$ layers, leading to the structural phase transition to the different layered structures of $P\bar{3}m1$ and $P\bar{3}1m$[29,30] depending on the hydrogen concentration at octahedral and tetrahedral sites as shown in Fig. 1b, c (see also Supplementary Fig. 1).

The $P\bar{3}m1$ structure (No. 164) is derived from the occupancy of 2 moles of hydrogen at only tetrahedral sites and the $P\bar{3}1m$ structure (No. 162) is derived from the excess occupancy at octahedral sites with additional one mole of hydrogen. Thus, hydrogens can substitute the IQEs at octahedral sites within the concentration of one mole, maintaining the pristine $R\bar{3}m$ structure. This is the reminiscent of the chlorine-substituted $Gd_2CCl$, where Cl anions only occupied the octahedral sites and the remained IAEs are fully delocalized in the enlarged interlayer space (see the comparison of structural parameters in Supplementary Fig. 1 and Supplementary Table 1), crystallizing into the same $R\bar{3}m$ structure[11]. Furthermore, as the $Gd_2CCl_2$ is crystallized into the $P\bar{3}m1$ structure with the occupancy of Cl atoms at the only tetrahedral sites[31], our total energy calculations expect that the $Gd_2CH_2$ are also crystallized into the $P\bar{3}m1$ structure, exhibiting the preferential occupancy of hydrogens at the only tetrahedral sites (Supplementary Fig. 1b). It is also confirmed that the hydrogenated $Gd_2CH_3$ with the excess occupancy at octahedral sites is stabilized in the $P\bar{3}1m$ structure (Supplementary Fig. 1c).

Pristine $[Gd_2C]^{2+}\cdot 2e^-$ electride was annealed at 600 K and 1000 K under hydrogen pressure of $10^{-1}$ Torr with 4% $H_2$ mixed Ar gas (Supplementary Fig. 2). In order to ensure a homogeneity of hydrogen distribution in the samples, the pulverized powders of the electride were annealed for 24 h. The hydrogenated samples were investigated by X-ray diffraction (XRD) pattern measurement and analyzed by Rietveld refinement method. The powder XRD pattern of the sample hydrogenated at 1000 K (hereafter referred as $Gd_2CH_x$ ($x\le 1.0$)) shows a good consistency with the simulated pattern of $R\bar{3}m$ structured pristine $[Gd_2C]^{2+}\cdot 2e^-$ electride (Fig. 1d). On the other hand, the sample hydrogenated at 600 K clearly showed a structural transition to $P\bar{3}1m$ structure (Fig. 1e, f). The XRD pattern of the sample hydrogenated at 600 K was well refined with the $P\bar{3}1m$ structure rather than $P\bar{3}m1$ structure (see Supplementary Table 2 for the comparison of Rietveld refinement results with $P\bar{3}m1$ and $P\bar{3}1m$ and Supplementary Fig. 3 for the refinement results with $P\bar{3}1m$ structure of another sample).

This structural phase transition is reminecent of the hydrogenation of the isostructural $[Y_2C]^{2+}\cdot 2e^-$ electride, yielding the $P\bar{3}1m$ structure of $Y_2CH_{2.55}$, where 2 moles of hydrogen occupy all tetrahedral sites and additional 0.55 moles of hydrogen occupy the octahedral sites, as confirmed from neutron diffraction (ND) study[29,30]. Our total energy calculations on the $Gd_2CH_3$ compound, which showed the full occupancy at both octahedral and tetrahedral sites, is consistent to the $P\bar{3}1m$ structure of $Y_2CH_{2.55}$ (Supplementary Fig. 1c). Although the ND measurements of hydrogenated $Gd_2CH_x$ and $Gd_2CH_y$ compounds are not possible due to a high absorption nature of neutron beam by Gd atoms, our hydrogenation condition was verified by the hydrogenation of the isostructural $[Y_2C]^{2+}\cdot 2e^-$ electride and its ND measurements followed by Rietveld refinements, ensuring the hydrogen concentration over 2 moles and structural phase transition to $P\bar{3}1m$ of $Gd_2CH_y$ ($y>2.0$) for the sample hydrogenated at 600 K (Supplementary Fig. 4). Crystal structure data derived from the Rietveld refinements of XRD and ND measurements are given in Supplementary Tables 2 and 3.

## Hydrogen-induced magnetic transition in $[Gd_2C]^{2+}\cdot 2e^-$ electride

The IAEs have played a key role in governing the electronic and magnetic behavior of the $[Gd_2C]^{2+}\cdot 2e^-$ electride, showing the anisotropy in the metallic conduction and magnetic property. In particular, the ferromagnetism of the $[Gd_2C]^{2+}\cdot 2e^-$ electride was explained from the strongly localized IAEs with their own magnetic moments, which facilitate the exchange interaction between Gd–Gd atoms across the IAEs in the interlayer space[11]. This quasi-atomic nature of the IAEs can

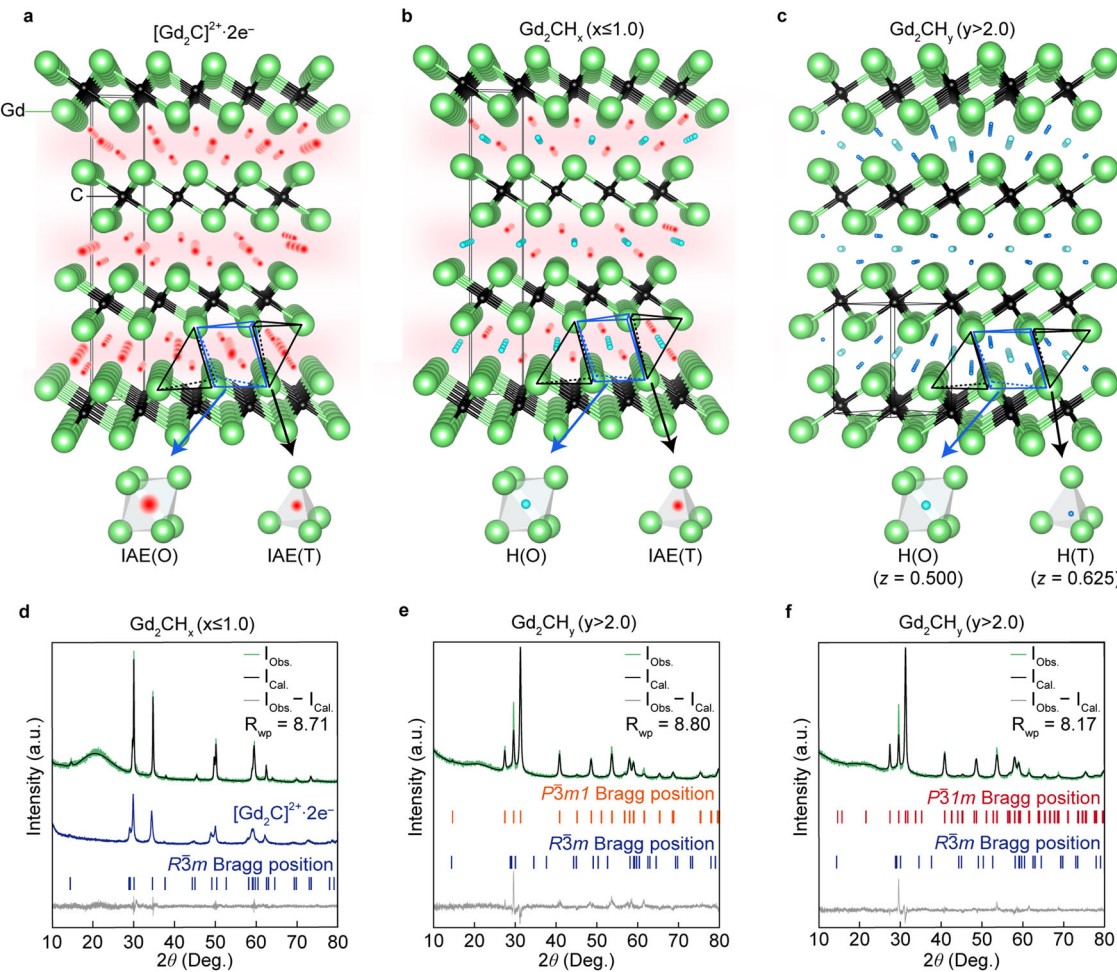

**Fig. 1 | Hydrogen-induced structural phase transition in $[Gd_2C]^{2+}\cdot2e^-$ electride and its hydrides. a–c** Crystal structure of $[Gd_2C]^{2+}\cdot2e^-$ (**a**), $Gd_2CH_x$ ($x \leq 1.0$) (**b**), and $Gd_2CH_y$ ($y > 2.0$) (**c**), where the IAEs (red) and hydrogens (blue) occupy octahedral (O) and tetrahedral (T) sites of interlayer space between Gd cationic layers. The Wyckoff positions of IAEs and hydrogens are derived from the previous reports[29,30] and powder ND analysis of hydrogenated $[Y_2C]^{2+}\cdot2e^-$ (Supplementary Fig. 4). **d** –**f** Rietveld refinement of powder XRD patterns for $Gd_2CH_x$ ($x \leq 1.0$) (**d**) and $Gd_2CH_y$ ($y > 2.0$) (**e**, **f**). Bragg position of the pristine $R\bar{3}m$ structured $[Gd_2C]^{2+}\cdot2e^-$ is shown for a comparison in **d**. The $P\bar{3}m1$ structure (No. 164) and $P\bar{3}1m$ structure (No. 162) were used for the refinements of $Gd_2CH_y$ ($y > 2.0$) in **e** and **f**, respectively. Source data are provided as a Source Data file.

be verified by the substitution with other elements and subsequent change in electrical and magnetic properties. Figure 2a shows the temperature dependence of electrical resistivity for the pristine $[Gd_2C]^{2+}\cdot2e^-$ electride and hydrogenated samples. The normalized resistance, $R/R_{400K}$ of the $[Gd_2C]^{2+}\cdot2e^-$ electride and the hydrogenated $Gd_2CH_x$ ($x \leq 1.0$) decreased with the decrease in temperature, showing the same metallic behavior. The increased $R/R_{400K}$ and enhanced electron-phonon scattering in the hydrogenated $Gd_2CH_x$ ($x \leq 1.0$) imply the reduced concentration of the IQEs (Supplementary Fig. 5). However, the hydrogenated $Gd_2CH_y$ ($y > 2.0$) showed the typical behavior of semiconductors, indicating that most IQEs are substituted by the hydrogens and metal-insulating transition occurs according to their concentration.

A profound effect of hydrogenation was found in the magnetic properties. Figure 2b shows the temperature dependence of magnetic susceptibility ($\chi$) for the pristine $[Gd_2C]^{2+}\cdot2e^-$ electride and its hydrogenated samples. Compared to the ferromagnetic transition temperature ($T_C$) of 350 K of the $[Gd_2C]^{2+}\cdot2e^-$ electride, the hydrogenated $Gd_2CH_x$ ($x \leq 1.0$) sample exhibits the decreased $T_C$ of 220 K, which was determined from the temperature dependence of $dM/dT$ curve (Supplementary Fig. 6a). This $T_C$ decrease is obviously attributed to the reduced IQE concentration by the substitution with hydrogen atoms. Furthermore, this behavior is reminiscent of well-known strategy for

tuning the $T_C$ of ferromagnets by the substitution with non-magnetic elements[23], proving the magnetic nature of the IQEs. An interesting feature is the enhanced $\chi$ and the bifurcation between ZFC and FC curves in the hydrogenated $Gd_2CH_x$ ($x \leq 1.0$) sample. This hydrogenated $Gd_2CH_x$ ($x \leq 1.0$) is clearly distinguished from the antiferromagnetic $Gd_2CCl$, where the Cl anions occupy the only octahedral sites and lead to the substantial increase of the interlayer distance (0.414 nm) that is even larger than that (0.386 nm) of the $[Ca_2N]^+\cdot e^-$ electride with fully delocalized IAEs (Supplementary Fig. 7). Thus, the remained IAEs in the $Gd_2CCl$ can be fully delocalized at the interlayer, hardly affecting the exchange interaction by removing the localized nature of IQEs. In contrast, the $Gd_2CH$ shows the decreased interlayer distance of 0.314 nm than that (0.338 nm) of the pristine $[Gd_2C]^{2+}\cdot2e^-$ electride[11], probably resulting in much stronger localization of the remained IAEs at both octahedral and tetrahedral sites and reinforcing the ferromagnetic Gd-IQEs-Gd exchange interaction in the hydrogenated $Gd_2CH_x$ ($x \leq 1.0$) sample. This ascribes to the enhanced $\chi$ and the random distribution of the IAEs at the octahedral sites may induce the bifurcation between ZFC and FC curves. Meanwhile, the role of hydrogen anions at the octahedral sites in the $Gd_2CH$ compound is worthy to be further investigated whether the hydrogen anions between 2D Gd arrays can impart a ferromagnetic exchange interaction.

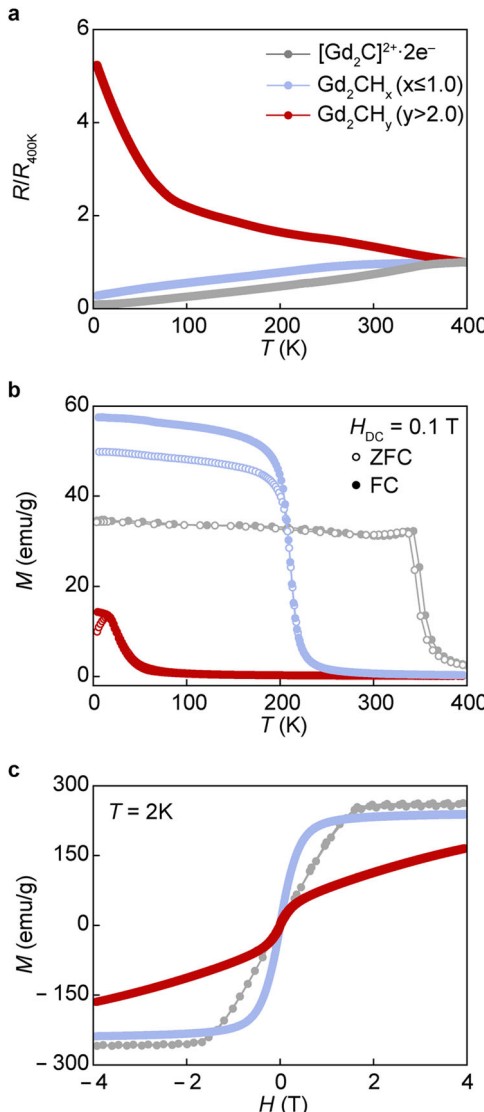

**Fig. 2 | Hydrogen-induced transitions of electrical and magnetic properties in [Gd₂C]²⁺·2e⁻ electrode and its hydrides.** **a** Temperature dependence of normalized resistance, $R/R_{400K}$, for pristine $[Gd_2C]^{2+}·2e^-$ and hydrogenated $Gd_2CH_x$ ($x \leq 1.0$) and $Gd_2CH_y$ ($y > 2.0$) samples. **b** Temperature dependence of magnetization ($M$) measured under 0.1 T for the three samples. **c** Magnetic field ($H$) dependence of $M$ measured at 2 K for the three samples. Source data are provided as a Source Data file.

Importantly, it should be noted that the magnetic transition occurs by the substitution of most IQEs with hydrogens. The $M$–$T$ curve in the $P\bar{3}1m$ structured $Gd_2CH_y$ ($y > 2.0$) exhibits a typical feature of antiferromagnetic materials with ZFC downturn behavior. The isothermal magnetization ($M$–$H$ curve) was also measured at 2 K. No hysteresis loop was observed in every sample, discarding a hard ferromagnetic ground state. Although the $M$–$H$ curves exhibited the saturation of $M$ at small $H$ in the $R\bar{3}m$ structured $[Gd_2C]^{2+}·2e^-$ electrode and hydrogenated $Gd_2CH_x$ ($x \leq 1.0$) sample, the $P\bar{3}1m$ structured $Gd_2CH_y$ ($y > 2.0$) showed no saturation even at 5 T, suggesting an antiferromagnetic ground state. In the Curie-Weiss (CW) analysis on the inverse susceptibility curve ($\chi^{-1}$ vs. $T$) for all the samples (Supplementary Fig. 6b), the $[Gd_2C]^{2+}·2e^-$ electrode and hydrogenated $Gd_2CH_x$ ($x \leq 1.0$) sample followed the CW fit. However, a slight deviation from the linear fit was observed in the hydrogenated $Gd_2CH_y$ ($y > 2.0$) sample, suggesting the antiferromagnetic nature.

## Canted antiferromagnetism induced by hydrogenation

To clearly understand the antiferromagnetic properties of the hydrogenated $Gd_2CH_y$ ($y > 2.0$), we further examined AC susceptibility. Figure 3 shows the results of AC susceptibility measurements under 1.5 Oe of $H_{AC}$ with different frequencies. Both the real part ($\chi'$) and imaginary part ($\chi''$) extracted from the AC susceptibility of ferromagnetic $[Gd_2C]^{2+}·2e^-$ and hydrogenated $Gd_2CH_x$ ($x \leq 1.0$) showed the constant temperature of peak position ($T_p$, indicated by arrow) regardless of frequency (Fig. 3a, b and Supplementary Fig. 8). On the other hand, both $T_p$ of the $\chi'$ and $\chi''$ for the hydrogenated $Gd_2CH_y$ ($y > 2.0$) showed a systematic shift towards higher temperatures with the increase in frequency (Fig. 3c, d and Supplementary Fig. 9). This increasing $T_p$ behavior and non-zero net magnetic moment of $\chi''$ confirms the canted spin ordering in the hydrogenated $Gd_2CH_y$ ($y > 2.0$).

In contrary to the antiferromagnetic $Gd_2CCl$, where the IQEs are substituted by Cl atoms and the 2D array of gadolinium cations was responsible for the magnetism without splitting between ZFC and FC curves[11], the $Gd_2CH_y$ ($y > 2.0$) demonstrates non-zero net magnetization and weak hysteresis in $M$–$H$ curve (Supplementary Fig. 10). These behaviors indicate that the exchange interaction of Gd–IQEs–Gd was largely suppressed by the substitution of IQEs with hydrogens, but the presence of hydrogen anions probably leads to the canted spin structure between the 2D gadolinium arrays. Critical behavior analysis around magnetic transition temperature provides a plausible canted antiferromagnetic structure of the hydrogenated $Gd_2CH_y$ ($y > 2.0$), where the XY model is not available below the $T_N$ (Supplementary Figs. 11 and 12f, i), indicating that the hydrogen anions trigger the antiferromagnetic spin canting for the out-of-plane Gd–Gd atoms across the non-magnetic hydrogen anions as shown in the schematic illustration of Supplementary Fig. 13.

## Reversible structural and magnetic phase transitions

A strong coupling between structural and magnetic phase transitions in the hydrogenation of the $[Gd_2C]^{2+}·2e^-$ electrode is also observed in the dehydrogenation of the hydrogenated $Gd_2CH_y$ ($y > 2.0$). Dehydrogenation was conducted by heating the hydrogenated $Gd_2CH_y$ ($y \geq 2.0$) under a vacuum of $10^{-5}$ Torr. Figure 4a shows the $M$–$T$ curves of the dehydrogenated samples of $Gd_2CH_y$ ($y > 2.0$) at different temperatures together with the $[Gd_2C]^{2+}·2e^-$ (gray) and its hydrogenated samples (blue and red). The decreased magnetic transition temperature upon the hydrogenation, from $T_C$ ~ 350 K of the $[Gd_2C]^{2+}·2e^-$ electrode to $T_N$ ~ 20 K of the hydrogenated $Gd_2CH_y$ ($y > 2.0$), is perfectly reversed by the dehydrogenation, which increases the magnetic transition temperatures with the increase in dehydrogenation temperature. Notably, the magnetic transition temperature increases up to $T_C$ ~ 350 K, which is exactly same as that of the pristine $[Gd_2C]^{2+}·2e^-$ electrode. It is also noted that the dehydrogenated samples show magnetic transition from antiferromagnetism (dehydrogenated at 500 K and 800 K) to ferromagnetism (dehydrogenated at 1000 K and 1500 K) with proceeding the dehydrogenation. This is the reverse of the hydrogenation of the $[Gd_2C]^{2+}·2e^-$ electrode accompanied by the coupled structural and magnetic phase transitions. Indeed, it was confirmed from the XRD measurements of dehydrogenated samples (Fig. 4b) that the structural transition occurs from the $P\bar{3}1m$ structure of hydrogenated $Gd_2CH_y$ ($y > 2.0$) and its dehydrogenated sample at 500 K to $R\bar{3}m$ structure of dehydrogenated samples at 800 K, 1100 K, and 1300 K. Most of all, the $T_C$ as well as $M$–$T$ curve of the dehydrogenated sample at 1300 K indicate that the $[Gd_2C]^{2+}·2e^-$ electrode is re-formed by the re-generated magnetic IQEs at the original Wyckoff positions, re-inducing the ferromagnetic Gd–IQEs–Gd exchange interactions (Fig. 4c).

## Desorption of hydrogens and conservation of IAEs

It is worthwhile to consider that the magnetic IQEs are inherently conserved from the dehydrogenation of the hydrogenated electrode.

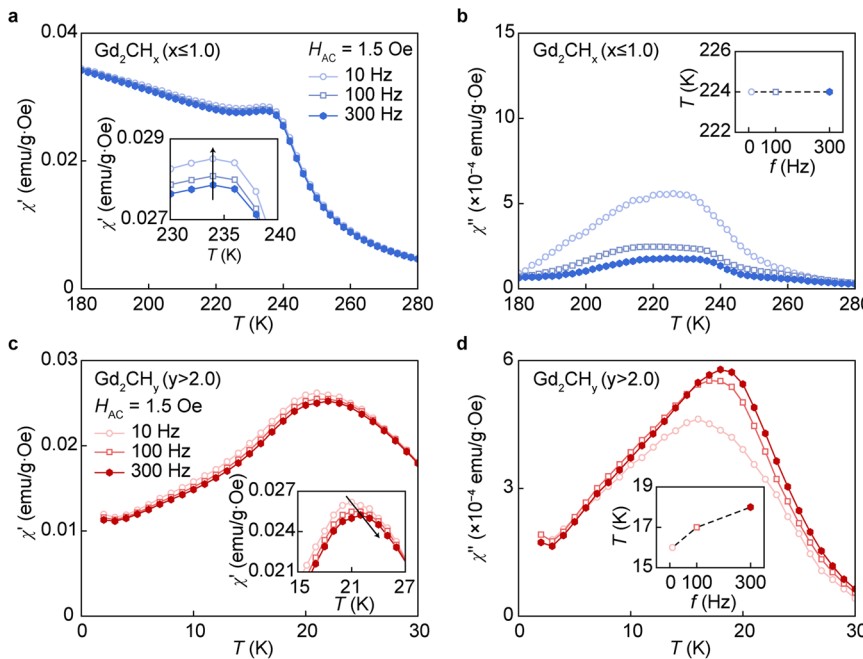

**Fig. 3 | AC susceptibility measurements for hydrogenated Gd₂CHₓ (x ≤ 1.0) and Gd₂CHᵧ (y > 2.0). a, b** Real part ($\chi'$) (**a**) and imaginary part ($\chi''$) (**b**) of AC susceptibility for Gd₂CHₓ (x ≤ 1.0). **c, d** $\chi'$ (**c**) and $\chi''$ (**d**) of AC susceptibility for Gd₂CHᵧ (y > 2.0). Insets in **a** and **c** show the constant and increasing behavior of the $\chi'$ near the phase transition points under different frequencies, respectively. Insets in **b** and **d** are the plot of maximum $\chi''$ depending on the frequency, showing the constant and increasing behavior, respectively. Source data are provided as a Source Data file.

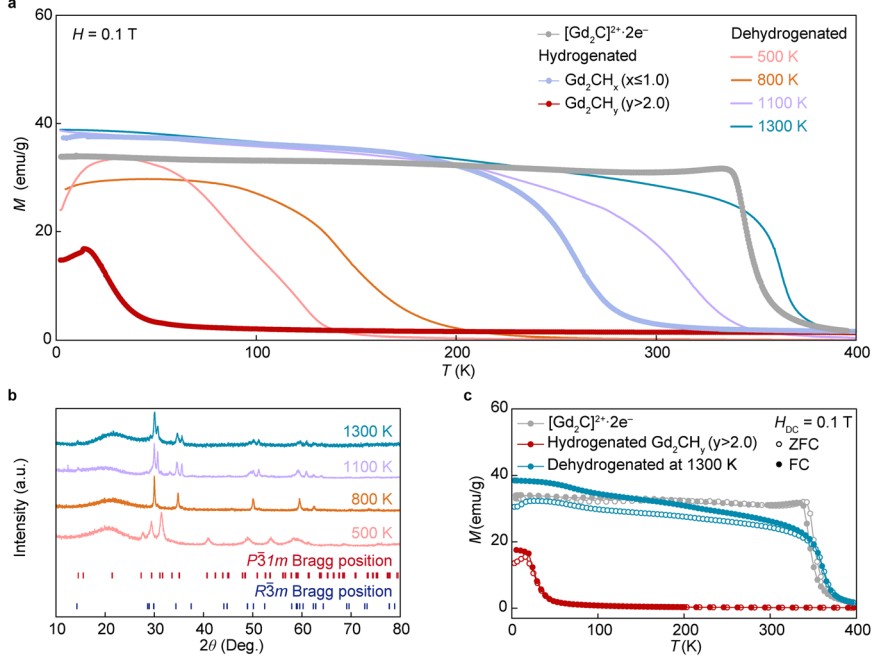

**Fig. 4 | Reversible structural and magnetic transitions between [Gd₂C]²⁺·2e⁻ electride and its hydrides. a** Temperature dependence of magnetization ($M$) for pristine [Gd₂C]²⁺·2e⁻, hydrogenated Gd₂CHₓ (x ≤ 1.0) and Gd₂CHᵧ (y > 2.0), and dehydrogenated samples of hydrogenated Gd₂CHᵧ (y > 2.0) at different temperatures. Magnetic transition temperatures obtained by Curie-Weiss fitting are shown in Supplementary Fig. 14. **b** Powder XRD patterns of dehydrogenated samples of hydrogenated Gd₂CHᵧ (y > 2.0) at different temperatures. **c** $M$–$T$ curves of pristine [Gd₂C]²⁺·2e⁻, hydrogenated Gd₂CHᵧ (y > 2.0) and sample dehydrogenated at 1300 K. $M$–$T$ curves are shown in Supplementary Fig. 15. Source data are provided as a Source Data file.

When both processes of hydrogenation and dehydrogenation were completed, the positively charged [Gd₂C]²⁺ layers in the $R\bar{3}m$ and $P\bar{3}1m$ structures were compensated by the IQEs and hydrogen anions, respectively. Furthermore, the charge neutrality of hydrogenated Gd₂CHₓ (x ≤ 1.0) and dehydrogenated samples at 500 K, 800 K, and 1100 K are also maintained by the coexistence of IQEs and hydrogen anions in the positively charged [Gd₂C]²⁺ layered lattice framework. These results indicate that the continuous substitution between IQEs

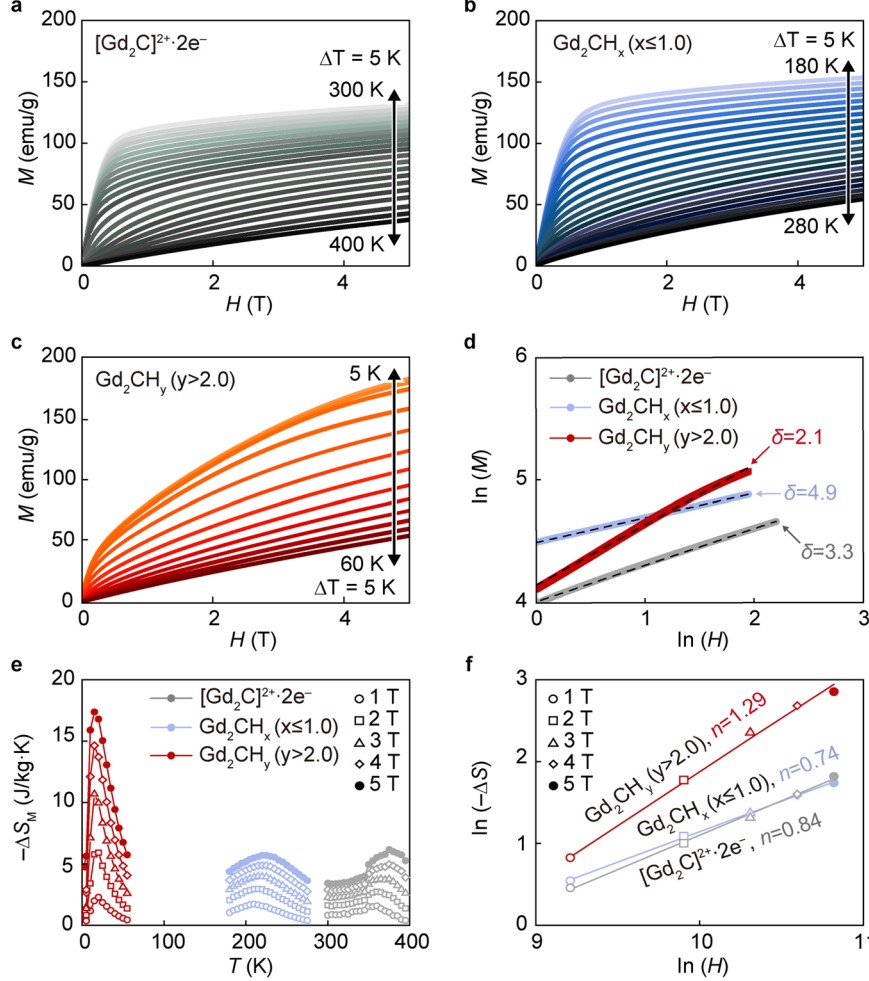

**Fig. 5 | Tunable magnetocaloric effect between [Gd$_2$C]$^{2+}$·2e$^-$ electride and its hydrides. a–c** Isothermal magnetization of pristine [Gd$_2$C]$^{2+}$·2e$^-$ electride (**a**), hydrogenated Gd$_2$CH$_x$ ($x \le 1.0$) (**b**), and Gd$_2$CH$_y$ ($y > 2.0$) (**c**). **d** ln($M$)−ln($H$) plot at each magnetic transition temperature for the three samples. **e** Difference in

magnetic entropy ($\Delta S_M$) under magnetic field of 1−5 T. **f** ln($-\Delta S_M$)−ln($H$) plot for the [Gd$_2$C]$^{2+}$·2e$^-$ electride and its hydrogenated samples. Source data are provided as a Source Data file.

with hydrogens occurs during the both processes. Although the reaction of IQEs with hydrogens, which produces the hydrogen anions in the hydrogenated samples, can be reasonably expected, the conservation of IQEs by the desorption of hydrogens in the dehydrogenated samples is, to the best of our knowledge, an unreported phenomenon when considering that the formation of vacancy at the site of hydrogen ions is a general feature in the dehydrogenated materials[32,33]. Many theoretical studies have explored the nature of electride by the substitution of IQEs with hydrogen anions and experimental results have been reported to provide evidence on the existence of the IQEs in the electrides[34,35]. However, to our best knowledge, the dehydrogenation producing the electride has never been reported yet. Indeed, proof-of-demonstrations on the magnetic IQEs of the electrides have been rare in experiments. In contrary to the well-known intercalation and deintercalation processes of elements in 2D materials[36,37], the reversible transitions depending on the stoichiometric balance between IQEs and hydrogen anions strongly indicates that the IAEs at a specific Wyckoff position behave as quasi-atoms to keep the charge neutrality and to ensure the thermodynamic and electronic stability of the electrides[11,19,38]. In particular, because the continuous change in magnetic transition temperatures is exclusively ascribed to the relative concentration of magnetic IQEs and hydrogens, the nature of IAEs can be regarded as magnetic IQEs, allowing the freedom to tune the magnetic properties of electrides.

## Magnetocaloric effect upon hydrogenation

A magnetocaloric effect (MCE) can also provide an insight into the nature of magnetic phase transition and the existence of magnetic IQEs, which are related to the difference in the degree of freedom of spin alignment. The MCE of a system with a weaker exchange interaction shows a larger magnetic entropy ($S_M$) near the magnetic transition temperature. The $\Delta S_M$ can be derived from isothermal magnetization as shown in Fig. 5a–c. These measurements give the nature of spin exchange interaction from the critical exponent $\delta$ value in the relation of $M(H,T) = DH^{1/\delta}$, ($T = T_C$), which can be obtained from the slope of ln ($H$) − ln ($M$) plot (Fig. 5d). The $\delta$ value of 2.1 for the canted antiferromagnetic hydrogenated Gd$_2$CH$_y$ ($y > 2.0$) is well-matched with the spin-disordered state[39–43]. For the [Gd$_2$C]$^{2+}$·2e$^-$ and hydrogenated Gd$_2$CH$_x$ ($x \le 1.0$), the $\delta$ values of 3.3 and 4.9 are of ferromagnetic systems. Furthermore, the temperature dependence of the $\Delta S_M$ from the following equation,

$$\Delta S_M = \int_0^H \left( \frac{\partial M}{\partial T} \right)_H dH \qquad (1)$$

is shown in Fig. 5e. The isothermal $\Delta S_M$ showed a broad maximum around $T_C$ for the ferromagnetic [Gd$_2$C]$^{2+}$·2e$^-$ and Gd$_2$CH$_x$ ($x \le 1.0$), whereas the canted antiferromagnetic Gd$_2$CH$_y$ ($y > 2.0$) exhibited a sharp increase around $T_N$. The maximum entropy change at $T_C$ was

found to be as high as 17.3 J·kg$^{-1}$K$^{-1}$ for the Gd$_2$CH$_y$ ($y > 2.0$), which is almost three times higher than those of [Gd$_2$C]$^{2+}$·2e$^-$ and Gd$_2$CH$_x$ ($x \le 1.0$) samples, indicating that a weaker exchange interaction is present for the hydrogenated Gd$_2$CH$_y$ ($y > 2.0$). In addition, the power-law fitting of $-\Delta S_M \propto H^n$ (Fig. 5f) also implies that the hydrogenated Gd$_2$CH$_y$ ($y > 2.0$) with $n$ greater than 1.0 follows the antiferromagnetic behavior, which is clearly distinguished from the ferromagnetic [Gd$_2$C]$^{2+}$·2e$^-$ and Gd$_2$CH$_x$ ($x \le 1.0$). Besides of the identification of the change in the magnetism according to the concentration of magnetic IQEs, the magnetic properties such as relative cooling power (RCP) can be also controlled by the magnetic IQEs. We calculated the RCP with the equation of $|\Delta S_M^{max}| \times \delta T_{FWHM}$ (Supplementary Fig. 16), where the $\delta T_{FWHM}$ is the full width at half maxima of the peak of the $\Delta S_M$ versus $T$ plot in the Fig. 5e. Clearly, the RCP of hydrogenated Gd$_2$CH$_y$ ($y > 2.0$) sample is quite larger (~350 J·kg$^{-1}$ at 5 T) than those of the [Gd$_2$C]$^{2+}$·2e$^-$ and Gd$_2$CH$_x$ ($x \le 1.0$) samples, showing a comparable capability to that of magnetocaloric AlFe$_2$B, Gd$_5$Si$_4$ and Gd$_2$FeAlO$_6$ materials[44,45]. These exotic properties based on the magnetic IQEs can provide a possibility to explore a refrigerant from electride material.

## Discussion

In summary, we explored the non-magnetic hydrogen substitution for the magnetic IQEs in the 2D [Gd$_2$C]$^{2+}$·2e$^-$ electride and found the strongly coupled structural and magnetic phase transitions. This coupling was also observed by the dehydrogenation of the Gd$_2$CH$_y$ ($y > 2.0$), which perfectly conserved the IQEs. The structural phase transition between higher symmetric $R\bar{3}m$ structure of the [Gd$_2$C]$^{2+}$·2e$^-$ electride and lower symmetric $P\bar{3}1m$ structure of the hydrogenated Gd$_2$CH$_y$ ($y > 2.0$) is accompanied with the magnetic phase transition in a wide temperature range between ferromagnetism at the $T_C$ of 350 K to canted antiferromagnetism at the $T_N$ of 20 K. The reversible magnetic transition is governed by the spin exchange interactions in the out-of-plane Gd−Gd atoms, which are mediated across the magnetic IQEs or non-magnetic hydrogen anions. Our results clearly identified the nature of magnetic IQEs and proved their critical role in tuning the magnetic properties, providing the IQEs as a unique ingredient in magnetic materials. These IQEs, which can have interactions with each other or surrounding cations, can thus trigger antiferromagnet, ferromagnet, or permanent magnet, all the magnetism. Finally, the reversible substitution and conservation between magnetic IQEs and hydrogen anions can provide a possible platform to study the exotic magnetic state of quantum electron phases such as Wigner crystal on the electrides[22,46].

## Methods

### Synthesis of [Gd$_2$C]$^{2+}$·2e$^-$ electride and its hydrides

All samples were handled in glove boxes filled with high-purity argon gas (Ar 99.999%) to prevent the oxidation of raw materials and synthesized samples. The synthesis method of a polycrystalline ingot of [Gd$_2$C]$^{2+}$·2e$^-$ electrides is performed by the arc-melting process with mixed Gd metal pieces and graphite pieces in a 2:1 molar ratio under high-purity Ar atmosphere. Before synthesizing the hydrogenated sample, we pulverized the polycrystalline [Gd$_2$C]$^{2+}$·2e$^-$ electrides to powder and pelletize for handling. As displayed in Supplementary Fig. 2, hydrogenation is performed in the quartz tube furnace under Ar-based 4% H$_2$ mixed gas. To synthesize the Gd$_2$CH$_y$ ($y > 2.0$) composition, pelletized [Gd$_2$C]$^{2+}$·2e$^-$ electrides were heat treatment of around 600 K under a 1 atm environment. The Gd$_2$CH$_x$ ($x \le 1.0$) was synthesized around 1000 K and under 10$^{-1}$ Torr pressure, which is created by flowing the H$_2$-mixed gas and vacuum pumping at the same time. The dehydrogenation process was performed by pelletized Gd$_2$CH$_y$ ($y > 2.0$) under 10$^{-5}$ Torr by vacuum pumping with different temperatures.

### Structural characterization by X-ray and neutron powder diffraction

The crystal structure of the [Gd$_2$C]$^{2+}$·2e$^-$ electride, hydrogenated samples (Gd$_2$CH$_x$ ($x \le 1.0$) and Gd$_2$CH$_y$ ($y > 2.0$)), and dehydrogenated samples were investigated by XRD using a Rigaku SmartLab diffractometer with monochromatic Cu K$_\alpha$ radiation (8.04 keV) at room temperature. The well-ground powder samples were prepared in glove boxes and measured in a plastic dome-type stage filled with Ar gas to avoid oxidation during measurements. A high resolution neutron powder diffraction of hydrogenated Y$_2$CH$_y$ ($y > 2.0$) sample was measured at the HANARO, a research reactor of the Korea Atomic Energy Research Institute. The wavelength of the neutron beam of HRPD is $\lambda$ = 0.1834528 nm, and the measurement error of the lattice change rate using this beam is about ± 0.004%. General Structure Analysis System (GSAS) software package was applied to perform Rietveld refinement.

### Magnetic and electrical properties characterization

The sampling for resistivity and magnetic properties measurements were performed in the high-purity Ar-filled glove boxes. The temperature-dependent resistivity measurements were performed by the physical property measurement system (PPMS DynaCool, Quantum Design). The four-electrode is made by silver epoxy on the pelletized samples. After that, samples were covered with Apiezon N grease to block the oxidation during the sample loading to PPMS and measurements. The measurement of magnetic properties used a vibrating sample magnetometer (VSM, Quantum Design) and Squid magnetometer (MPMS3, Quantum Design) for AC magnetic susceptibility. A plastic capsule copula containing a weighted sample was coated with N grease to prevent the oxidation of samples.

### Reporting summary

Further information on research design is available in the Nature Portfolio Reporting Summary linked to this article.

## Data availability

The data that support the plots within this paper and other findings of this study are available from the corresponding author upon request. Source data are provided with this paper.

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

## Acknowledgements

This work was supported by the National Research Foundation of Korea (NRF) grant funded by the Korea government (MSIT) (2022M3H4A1A01010832, 2022M3H4A1A01010829, 2022R1A2C2005210), and Samsung Science and Technology Foundation under project number SSTF-BA2101-04.

## Author contributions

S.W.K. designed and directed the project. S.Y.L and D.C.L. synthesized the electride and hydride samples, S.Y.L., D.C.L., and M.S.K. analyzed the crystal structure. S.Y.L., M.S.K., and J.Y.H. measured and analyzed magnetic properties. H.S.K measured powder ND patterns of hydrides. K.H.L. and S.W.K. co-wrote the manuscript with support from all authors.

## Competing interests

The authors declare no competing interests.
