## [Peer Review File · Nature Communications]

Magnetic quasi-atomic electrons driven reversible structural and magnetic transitions between electrider and its hydridesREVIEWER COMMENTS

Reviewer #1 (Remarks to the Author):

In their research, Kim et al. investigate the structural and magnetic transitions triggered by the hydrogenation of a Gd electride into hydrides, as well as the converse process. They discovered that these transitions are intimately connected and primarily influenced by the existence of anionic electrons in the electrides. Notably, they observed that hydrogenation incites canted antiferromagnetism.

The study offers additional experimental evidence underscoring the influence of interstitial "quasi-atomic" electrons (IQEs) on the magnetic properties of electrides. One of the most fascinating aspects of this research is the observation that the process can be entirely reversed, allowing the dehydrogenated hydride to retain the IQEs of the original electride.

To my knowledge, this work is the first to demonstrate the creation of an electride through direct dehydrogenation. This significant discovery paves the way for exploring other exotic magnetic materials, opening a new realm of potential investigations.

Reviewer #2 (Remarks to the Author):

The paper shows an interesting phenomenon that the electride properties and the magnetic properties undergo systematic changes under the inclusion of H into the interatomic layers. Especially the change is reversible, namely while the material is dehydrogenated, its structural and magnetic properties can be reversed back to a non- or low- hydrogenated state. The paper might make a good contribution to studying the 2D and layered electride materials, especially to the control and the understanding of the intricate interplay between the quasi-atoms and the magnetism. The paper can be further considered to be published in Nature Communications. However, there are several aspects of the work that needs attention and implementation.

1. The team conducted thorough and very demanding experiments on the hydrogenation and dehydrogenation of the samples and the measurement of the magnetic susceptibilities etc. On the other hand, DFT calculations can help to understand many aspects of the materials, such as the stability, the origin of the complex magnetic patterns and evolutions under different levels of hydrogenation, etc. Including a thorough DFT study can greatly enhance the readability and understanding of the work.

2. The relation and comparison between the current work (hydrogenation) and the previous work (halogenation) need to be explicitly discussed. Is it true that the major difference between the two systems is due to the fact that H and halogens will occupy different sites in between the layers? Again, DFT calculations can help as they can reveal the energy preferences of different absorption sites.

3. It might not be proper to emphasize that this procedure can be used to control the electronic and magnetic properties because the control of the H insertion is considerably a harder process than the direct control of the magnetism and electronic properties. However, including H can be used to modify the material's properties.

4. Labels in some figures are too small to read.

Reviewer #3 (Remarks to the Author):

This is an interesting paper that looks at the property of the $(\text{Gd}_2\text{C})_{+2} \cdot 2e^-$ electride as hydrogen is added or removed from the structure. There is a change in the crystal structure as well as a change in the magnetic ordering temperature. The most striking finding by the authors is that after the sample is completely hydrogenated to $\text{Gd}_2\text{CH}_{x>2}$, and the magnetic ordering temperature changes from 350 K down to 21 K, the hydrogen can be removed by heating in vacuum and to a good approximation the original electride properties are obtained and the magnetic ordering returns to near 350 K. In general the results in this paper are relatively clear and interesting. The only objection I have to this work is the claim that the fully hydrogenated sample is a canted antiferromagnet. There is some evidence of this in the ac susceptibility and magneto caloric measurements but it is not clear to me how strong the claim is. Neutron diffraction is difficult with Gd but not impossible. A neutron diffraction study would be very helpful to support or prove the claims of the authors about the ordered magnetic state and the degree of magnetism associated with the electron anions. I think that neutron diffraction experiments would be an interesting study for a future paper. As far as I can tell this paper is complete enough and suitable for publication in Nat. Comm. as written.

Responses to the reviewers' comments

Reviewer #1 (Remarks to the Author):

In their research, Kim et al. investigate the structural and magnetic transitions triggered by the hydrogenation of a Gd electride into hydrides, as well as the converse process. They discovered that these transitions are intimately connected and primarily influenced by the existence of anionic electrons in the electrides. Notably, they observed that hydrogenation incites canted antiferromagnetism. The study offers additional experimental evidence underscoring the influence of interstitial "quasi- atomic" electrons (IQES) on the magnetic properties of electrides. One of the most fascinating aspects of this research is the observation that the process can be entirely reversed, allowing the dehydrogenated hydride to retain the IQES of the original electride. To my knowledge, this work is the first to demonstrate the creation of an electride through direct dehydrogenation. This significant discovery paves the way for exploring other exotic magnetic materials, opening a new realm of potential investigations.

> We would like to deeply thank the reviewer for acknowledging the importance of our present work. We are particularly grateful for the reviewer's precise recognition of the reversibility between quasi-atomic electrons and hydrogen anions in electride and its hydride. This phenomenon can give an interesting question on which came first, the positive lattice or the anionic electron like the curious question of "which came first, the chicken or the egg?". This exotic science can be disclosed by our ongoing research on the anionic electrons that may construct a quasi-lattice and show a permanent magnetism in an electride.

Again, thank you very much for your supporting evaluation.

Reviewer #2 (Remarks to the Author):

The paper shows an interesting phenomenon that the electride properties and the magnetic properties undergo systematic changes under the inclusion of H into the interatomic layers. Especially the change is reversible, namely while the material is dehydrogenated, its structural and magnetic properties can be reversed back to a non- or low- hydrogenated state. The paper might make a good contribution to studying the 2D and layered electride materials, especially to the control and the understanding of the intricate interplay between the quasi-atoms and the magnetism. The paper can be further considered to be published in Nature Communications. However, there are several aspects of the work that needs attention and implementation.

> First of all, we appreciate the reviewer supporting our manuscript and giving an important comment. We are particularly grateful to the reviewer for acknowledging our efforts on the contribution to the 2D electrides and on the understanding the magnetism originated from the quasi-atomic electrons.

We fully agree with your insight that DFT calculations can help to understand the reported phenomena related to the hydrogens in the magnetic electride. In particular, the thermodynamic stability of the hydrogenated compounds and identification of their magnetic structures that you suggested are highly valuable to study further. As you might expect, we have indeed performed the theoretical calculations even before we submitted the present manuscript.

We will show the theoretical calculation results that we have obtained up to now and will introduce a further study to explain why your valuable suggestions are more important than the present our results.

1. The team conducted thorough and very demanding experiments on the hydrogenation and dehydrogenation of the samples and the measurement of the magnetic susceptibilities etc. On the other hand, DFT calculations can help to understand many aspects of the materials, such as the stability, the origin of the complex magnetic patterns and evolutions under different levels of hydrogenation, etc. Including a thorough DFT study can greatly enhance the readability and understanding of the work.

> Thank you very much for this valuable comments.

Yes, the hydrogenation and dehydrogenation of the electrides have never been experimentally demonstrated, obviously highlighting the present experimental study verifying the reversible transitions of crystal structure and magnetic property between the Gd_2C electride and its hydrides, as well as triggering an extensive study on the hydrogenation and dehydrogenation of other electrides that will diversify the physical properties of the electride materials. Our experimental demonstrations can also give an interesting question to the research society of electrides, “if the crystal structure of the

hydrogenated compounds remains and the anionic electrons exist at the position of hydrogen anions even after the dehydrogenation, is it possible to have a polymorphic electride with different physical properties?”. This question can be answered by “Yes” from our ongoing research and we will report soon. In order to explain which and how an electride can have a polymorph by hydrogenation and subsequent dehydrogenation processes, the theoretical calculations for thermodynamic stability of polymorphic electrides and their hydrides would be very important and critical to expand the electride research for diverse classes of material and physical properties.

We have calculated the thermodynamic stability of the hydrogenated compounds with a different hydrogen concentration and different hydrogen crystallographic positions. Figure R1 shows the total energy of the Gd_2CH_x ($x = 1, 2,$ and 3), where hydrogens occupy the octahedral and/or tetrahedral sites at the interlayers. When the hydrogen is one mole, the octahedral sites are preferentially occupied by the hydrogens, showing no transition in crystal structure. It is interesting that the tetrahedral sites will be fully occupied by the 2 moles of hydrogen and the octahedral sites are empty, showing the phase transition from the $R\bar{3}m$ structure to $P\bar{3}m1$ structure (No. 164). When additional one mole of hydrogen ($x = 3$) is introduced, phase transition to the distorted the $P\bar{3}1m$ structure (No. 162) is more preferred with the excess occupancy at octahedral sites. These calculation results are added in the revised supplementary information file (Supplementary Fig. 1) and related discussions are present in the revised manuscript as followings.

Fig. R1. Total energy comparison depending on the hydrogen concentration and position of (a) Gd_2CH , (b) Gd_2CH_2 , and (c) Gd_2CH_3 , where the green, black, light blue, and dark blue balls represent Gd, C, H(O), and H(T), respectively. Gd_2CH with hydrogen occupation at octahedral sites is preferred (left). Gd_2CH_2 with the hydrogen occupancy at the only tetrahedral sites is preferred (right). Gd_2CH_3 with hydrogen occupancy at both octahedral and tetrahedral sites is preferred to be crystallized into the $P\bar{3}1m$ structure (right). Left structure of c is crystallized in the $P\bar{3}m1$ structure.

The phase transitions to the $P\bar{3}m1$ and $P\bar{3}1m$ structured hydrides should be further confirmed by the neutron diffraction measurements. Indeed, we have tried many times at the HANARO, a neutron facility of the KAERI in Korea, but failed due to the Gd having a high neutron absorption property. This provokes the addition of neutron diffraction measurements of fully hydrogenated Y_2CH_y (Supplementary Fig. 3 in the original Supplementary Information file), which is iso-structure of fully hydrogenated Gd_2CH_y . We are now in seeking a neutron facility (as well as Mössbauer spectroscopy and spin ARPES) for a collaboration on the magnetic structure of Gd_2C electride and its hydrides to clearly identify the critical role of hydrogen anions in dominating the magnetic phase transition. As you know very well, it is rather difficult to conclude the magnetic structure of canted antiferromagnetism by the only theoretical calculations, strongly suggesting an additional “smoking gun” experiment such as neutron diffraction measurement. Furthermore, it would be very time-consuming to match the magnetic structures of both theoretical calculations and experiments. Another great difficulty is the precise measurement of the hydrogen concentration in the hydrogenated/dehydrogenated samples, which is a critical factor to calculate the electronic and magnetic structures. Considering the quite competitive situation on the research of hydrogenated electrides (*Chem. Sci.*, **7**, 4036–4043 (2016), *Sci. Adv.* **7**, eabe6819 (2021), *Phys. Rev. B.* **106**, 045138 (2022), *Angew. Chem. Int. Ed.* **61**, e202209187 (2022)), we sincerely ask to understand that our present work needs to be timely published, and the suggested comments would be investigated in a separated work.

Most of all, we assure that our theoretical calculations on the hydrides are undergoing, and your suggestions are more valuable than the present work. Figure R2 shows the calculated density of state and electronic band structures of the Gd_2CH , Gd_2CH_2 , and Gd_2CH_3 compounds. Also, the observed electronic band structure via angle-resolved photoemission spectroscopy using the single crystalline Gd_2CH_y compound is shown in Fig. R3 (please find the difference from the band dispersion of the Gd_2C electride, published in *Nat. Mater.* **21**, 1269 (2022)). Though the electronic band calculations absolutely require further detailed calculations such as fat band analysis, it is clear that the observed band originated from the hydrogens in Gd_2CH_y compound. This can be a breakthrough in the condensed matter physics in the viewpoint of experimentally observed hydrogen band in electronic structure, which might come from the formation of the quasi-lattice of hydrogens in our Gd_2CH_y compound. In this context, your suggested DFT calculations are more important, and our calculation results should be present after intensive verification together with the concrete experimental results of magnetic structure verification via neutron diffraction measurements and precise measurement of hydrogen concentration via nuclear reaction analysis/thermal desorption spectroscopy.

We deeply appreciate your kind understanding in advance.

Fig. R2. Electronic band structure and density of state of Gd_2C electride and Gd_2CH , Gd_2CH_2 , and Gd_2CH_3 compounds.

Fermi surface & $\Gamma - \text{M}$, $\Gamma - \text{K}$ dispersions

Fig. R3. Fermi surface and band dispersion of Gd_2CH_y compound. The cone-shaped band at Γ is clearly different from the V-shaped band in Gd_2C electride, which is recently reported in *Nat. Mater.* **21**, 1269 (2022).

2. The relation and comparison between the current work (hydrogenation) and the previous work (halogenation) need to be explicitly discussed. Is it true that the major difference between the two systems is due to the fact that H and halogens will occupy different sites in between the layers? Again, DFT calculations can help as they can reveal the energy preferences of different absorption sites.

> Thank you very much for the comments that make our manuscript to be more informative.

Distinct differences between the hydrogenated and halogenated (Cl) Gd₂C compounds are as follows:

(1) The reversible structural and magnetic transitions occur in the only hydrogenated compounds. In our trial experiments on the dehalogenation of the halogenated compounds (tried by the same procedure as the dehydrogenation process of the present work), we could not observe such reversible transition, giving a decomposition at high temperatures above 800 K.

(2) We haven't found the higher Cl concentration over 2 moles in our experiments and literatures, whereas the hydrogens can be incorporated into the Gd₂C electride over 2 moles, possibly up to 3 moles, having the $P\bar{3}1m$ structure (No. 162).

(3) When the hydrogen and Cl is one mole in the Gd₂C, respectively, the magnetism of the Gd₂CH can be ferromagnetic with a lower T_C than 350 K as the [Gd₂C]²⁺·□·1e⁻ (□ is vacant, as reported in *Nat. Comm.* **11**, Article number: 1526 (2020)), but the Gd₂CCl was antiferromagnetic.

On the other hand, similarity between hydrogenated and halogenated (Cl) Gd₂C compounds can be found in their crystallographic positions. As you can find in our previous work (*Nat. Comm.* **11**, Article number: 1526 (2020)), the Cl in the Gd₂CCl occupies the octahedral sites in the interlayer as the hydrogens in the Gd₂CH. Therefore, both systems have the same $R\bar{3}m$ structure. Also, both hydrogen and chlorine in the Gd₂CH₂ and Gd₂CCl₂ occupy the only tetrahedral sites, leaving the octahedral sites to be empty and constructing the same $P\bar{3}m1$ structure (No. 164). Figures R4 and Tables R1,R2 show the crystal structures and the details of the structural parameters including the crystallographic positions of hydrogen and chlorine. These are added in the revised supplementary information file and the related discussions are also added in the revised manuscript.

Although the hydrogen and chlorine are located at the same positions in the Gd₂CH and Gd₂CCl, they exhibit different magnetic properties. In our previous research, we demonstrated that the substitution of paramagnetic Cl for ferromagnetic IAEs at the octahedral sites in the Gd₂CCl ([Gd₂C]²⁺·1Cl⁻·1e⁻) leads to the fully delocalized state of the remained IAEs in the increased interlayer spacing (Fig. R4c and Table R1). In other words, Cl atoms kills the ferromagnetic Gd-IAE-Gd exchange interaction by increasing the distance between Gd-Gd interlayers due to their large ionic size and removing the localized IAEs at the tetrahedral sites. However, both Gd₂CH ([Gd₂C]²⁺·1H⁻·1e⁻) and [Gd₂C]²⁺·□·1e⁻ (Supplementary Fig. 7 in *Nat. Comm.* **11**, Article number: 1526 (2020)), which have much shorter interlayer space than that of Gd₂CCl and have the localized IAEs at the tetrahedral sites, remains as a ferromagnet. In addition, the hydrogen anions at the octahedral sites in the Gd₂CH ([Gd₂C]²⁺·1H⁻·1e⁻) may have a possibility to play a role to give an Gd-H⁻-Gd

exchange interaction, which should be identified in a further study based on the combined theoretical and experimental investigation.

These similarity and difference between hydrogenated and halogenated Gd_2C compounds are newly added in the introduction, result, and discussion parts with the red highlight, respectively.

Fig. R4. Comparison of hydrogenated ((a) Gd_2CH and (b) Gd_2CH_2) and halogenated ((c) Gd_2CCl and (d) Gd_2CCl_2) compounds. Green, black, light blue, dark blue and purple balls represent Gd, C, H(O), H(T), and Cl atoms, respectively.

Table R1. Crystal structure and structural parameters of Gd₂CH and Gd₂CCl.

Sample	Gd ₂ CH	Gd ₂ CCl
Space group	R $\bar{3}$ m	R $\bar{3}$ m
a, b (Å)	3.64	3.69
c (Å)	18.13	20.35
Volume (Å ³)	207.55	240.24

Atom	Position / Occupancy	
Gd1	(0 0 0.257) / 1.0	(0 0 0.268) / 1.0
Gd2		
C1	(0 0 0) / 1.0	(0 0 0) / 1.0
C2	(0.333 0.667 0) / 1.0	(0.333 0.667 0) / 1.0
H/Cl	(0 0 0.5)	(0 0 0.5)

Table R2. Crystal structure and structural parameters of Gd₂CH₂ and Gd₂CCl₂.

Sample	Gd ₂ CH ₂	Gd ₂ CCl ₂
Space group	P $\bar{3}$ m	P $\bar{3}$ m
a, b (Å)	3.72	3.76
c (Å)	6.12	9.46
Volume (Å ³)	73.42	116.02

Atom	Position / Occupancy	
Gd1	(0 0 0.257) / 1.0	(0.333 0.667 0.355) / 1.0
Gd2		
C1	(0 0 0) / 1.0	(0 0 0.5) / 1.0
C2	(0.333 0.667 0) / 1.0	
H/Cl	(0 0 0.257)	(0.333 0.667 0.834)

3. It might not be proper to emphasize that this procedure can be used to control the electronic and magnetic properties because the control of the H insertion is considerably a harder process than the direct control of the magnetism and electronic properties. However, including H can be used to modify the material's properties.

> We thank the reviewer for the comments. Yes, you are absolutely right.

As you know, the direct control of the magnetic and electronic properties for the electrides is very limited due to the nature of the electrides. Distinguished from the conventional materials engineering, which can generally adopt elemental substitution, alloying and doping, a strict regulation on the stoichiometry of chemical composition of electrides, which is essential to maintain the charge neutrality, makes difficult to control the materials properties of electrides. Furthermore, the physical and chemical properties of electrides are largely depending on the concentration, localization degree, dimensional array of anionic electrons. Indeed, theoretical studies have been mainly reported for the modulation of magnetic properties of 2D electrides (*J. Phys. Chem. C* **123**, 24698 (2019), *Phys. Rev. B.* **106**, 045138 (2022)), rather than experimental reports.

Nonetheless, we have succeeded in experimentally modulating the magnetic properties of electrides, which were allowed by the substitution of the cations with the only elements having the same valence state to the cations in pristine Y_2C and Gd_2C electrides such as Y_2C - Sc_2C solid solution (*J. Am. Chem. Soc.* **139**, 17277 (2017)) and Gd_2C - Y_2C solid solution (*npj Quantum. Mater.* **6**, Article number: 21 (2021)). However, the magnetic properties of Y_2C - Sc_2C solid solutions were hardly changed from the superparamagnetism of Y_2C electride. The Gd_2C - Y_2C solutions showed the ferrimagnetism when the composition ratio of Gd to Y was 1 to 1 and other compositions showed the ferromagnetism with a systematic change of T_c . Importantly, the experimental demonstration for the density control of the anionic electrons has been lack. Thus, our present hydrogenation/dehydrogenation can suggest a simple methodology to change the magnetic properties of the electrides via the density control of the anionic electrons.

As you might think, however, the precise control of the hydrogen concentration is harder and is necessary to be solved. Though this precise control should be done in a further study, we have obtained the concentration of hydrogens in the hydrogenated Gd_2CH_x ($x \leq 1.0$) and Gd_2CH_y ($y > 2.0$) samples by using the thermal desorption spectroscopy (TDS), showing the rather broad range of x value from 0.3 to 0.5 and y value from 2.0 to 2.5, as shown in Fig. R5 (only shown here for your information). It is very important to control and measure the precise concentration of hydrogens in the hydrogenated samples for your suggested theoretical calculations. We are planning the measurements of nuclear reaction analysis (NRA), which gives a very narrow detection range of hydrogen concentration in a sample (please refer <https://doi.org/10.1016/j.surfrep.2014.08.002>), enabling an accurate and reliable theoretical study as you suggested.

Fig. R5. Thermal desorption spectroscopy of the hydrogenated Gd_2CH_x ($x \leq 1.0$) and Gd_2CH_y ($y > 2.0$) samples.

Finally, we again appreciate your comments and carefully checked the manuscript to remove the exaggerated expressions.

4. Labels in some figures are too small to read.

> We thank the reviewer for the comments. We carefully checked the labels and symbols in the figures and revised them for a better visualization. Most of them in original figures are enlarged.

Reviewer #3 (Remarks to the Author):

This is an interesting paper that looks at the property of the $(\text{Gd}_2\text{C})^{+2}\cdot 2\text{e}^-$ electride as hydrogen is added or removed from the structure. There is a change in the crystal structure as well as a change in the magnetic ordering temperature. The most striking finding by the authors is that after the sample is completely hydrogenated to Gd_2CH_x $x > 2$, and the magnetic ordering temperature changes from 350 K down to 21 K, the hydrogen can be removed by heating in vacuum and to a good approximation the original electride properties are obtained and the magnetic ordering returns to near 350 K. In general the results in this paper are relatively clear and interesting. The only objection I have to this work is the claim that the fully hydrogenated sample is a canted antiferromagnet. There is some evidence of this in the ac susceptibility and magneto caloric measurements but it is not clear to me how strong the claim is. Neutron diffraction is difficult with Gd but not impossible. A neutron diffraction study would be very helpful to support or prove the claims of the authors about the ordered magnetic state and the degree of magnetism associated with the electron anions. I think that neutron diffraction experiments would be an interesting study for a future paper. As far as I can tell this paper is complete enough and suitable for publication in Nat. Comm. as written.

> We would like to express our gratitude to the reviewer for acknowledging the significance of our current work and providing comments for future work.

First of all, we totally agree with your insight for more clear identification on the canted antiferromagnetism. Indeed, we have tried a lot to get a reliable data for crystal and magnetic structure analysis at the HANARO, a neutron facility of the KAERI in Korea, but failed due to the Gd having a high neutron absorption property. This provokes the addition of neutron diffraction measurements of fully hydrogenated Y_2CH_y (Revised Supplementary Fig. 4), which is iso-structure of fully hydrogenated Gd_2CH_y . Nonetheless, the magnetic structure of canted antiferromagnetic Gd_2CH_y can be further solved by a neutron diffraction measurement as well as Mössbauer spectroscopy. We are now in seeking a neutron facility for a collaboration on the magnetic structure of Gd_2C electride and its hydrides to clarify the role of IAEs and hydrogen anions in dominating their exotic magnetism. Furthermore, theoretical study combined with different experiments (in particular for measuring the precise hydrogen concentration) is now ongoing and will be reported when a reliable result is obtained.

We assure that a clearer identification on the hydrogen-induced magnetic phase transition and more interesting magnetic property such as a permanent magnetism will be reported soon elsewhere.

Again, thank you very much for your supportive review and valuable comments.

REVIEWERS' COMMENTS

Reviewer #1 (Remarks to the Author):

I was not requesting additional review and have no further comments about the material the authors have added to the manuscript or the comments done by the other reviewers. As far as I am concerned, the manuscript can be published as is.

Reviewer #2 (Remarks to the Author):

The previous comments have all been addressed by the authors in great detail and depth. The new version is ready to be published.

Reviewer #3 (Remarks to the Author):

The authors have suitably addressed my initial comments concerning the research. I feel that this paper is now suitable for publication in Nature Communications.

Response to Referees Letter

REVIEWERS' COMMENTS

Reviewer #1 (Remarks to the Author):

I was not requesting additional review and have no further comments about the material the authors have added to the manuscript or the comments done by the other reviewers. As far as I am concerned, the manuscript can be published as is.

We would like to express our gratitude to the reviewer for acknowledging the significance of our current work.

Reviewer #2 (Remarks to the Author):

The previous comments have all been addressed by the authors in great detail and depth. The new version is ready to be published.

We deeply thank you very much for your kind concern.

Reviewer #3 (Remarks to the Author):

The authors have suitably addressed my initial comments concerning the research. I feel that this paper is now suitable for publication in Nature Communications.

Again, we thank you very much for your supporting evaluation.